# A need or a luxury? Parents' attitudes towards international schools from a linguistic perspective

Fahad Almulhim[1]*, Hamza Alshenqeeti[2], Mohammad Almoaily[3], Ali Alsaawi[4], Nesreen Alahmadi[2]

1 English Department, College of Arts, King Faisal University, Hofuf, Saudi Arabia, 2 Department of Languages and Translation, College of Arts and Humanities, Taibah University, Medinah, Saudi Arabia, 3 English Department, College of Language Sciences, King Saud University, Riyadh, Saudi Arabia, 4 English Department, College of Education, Majmaah University, Majmaah, Saudi Arabia

☉ These authors contributed equally to this work.
* fkalmulhim@kfu.edu.sa

## Abstract

English is the language of instruction at international schools in Saudi Arabia, which have traditionally been attended by the children of expatriate professionals living in the country. In recent years, Saudi parents have begun to enrol their children in international schools, which is a novel phenomenon in the Saudi context. This study assessed parents' attitudes towards international schools from a linguistic perspective as well as their attitudes towards the potential impact on their children's first language – Arabic. Using a mixed-methods research design, both quantitative and qualitative data were collected from 76 parents using a 30-item questionnaire with a combination of open-ended and close-ended question. The findings indicate that Saudi parents choose international schools for a variety of reasons, including strong reputations in science education (72.4% of the sample), high quality curricula (80%), and the intensive English input required in various university specialisations and the labour market since Saudi Arabia (85%). Other reasons and concerns were also reported by some parents in the qualitative data. Saudi families returning from time spent in English-speaking countries choose international programmes for their monolingual children, either as a bridge before enrolment in public schools or as a long-term option. Despite the increasing popularity of international schools, Saudi parents nevertheless report concerns about the influence of international schools on children's first language acquisition.

## 1. Introduction

English has become a dominant language across the globe and is used as a lingua franca in business, commerce, medicine and higher education. It is the language of more than 85% of international organisations, the industrial technology, tourism and

**Data availability statement:** All relevant data are within the article and its Supporting information files.

**Funding:** Dr Fahad Almulhim thanks the Deanship of Scientific Research, Vice presidency for Graduate Studies and Scientific Research, King Faisal University, Saudi Arabia [KFU261298] for supporting this research. Dr Ali Alsaawi extends the appreciation to the Deanship of Postgraduate Studies & Scientific Research at Majmaah University for funding this research work through the project number (R-2026-98).

**Competing interests:** The authors have declared that no competing interests exist.

finance, and most research publications, conferences, technical terms and database interfaces [1]. The dominance of English has influenced approximately one billion students worldwide to learn the language [2]. In the last two decades, the number of English learners has grown, as indicated by a notable increase in the international programmes offered by primary, secondary and tertiary education.

Both industrialised and developing countries strive to prepare their younger generations with the skills needed to succeed in the global labour market, and so do educators in international schools [3]. One of these skills is competency in English. While Saudi students preferred Arabic-language programmes in the 1980s and 1990s [4,5], the prevalence of KSA universities now teaching an increasing number of programmes entirely in English suggests a shift. The continued use of Arabic has been linked to textbook availability [5]; however, many resources and textbooks from English-speaking authors intended for graduate-level studies are published and available in English. Policymakers thus have two options: translating and culturally adapting up-to-date textbooks or preparing students from an early age to achieve strong English skills. Many universities in Saudi Arabia now have a compulsory intensive preparatory (foundation) year that mandates studies in specific areas, including English [6], and both universities and recruiters appear to express a greater demand for more advanced levels of English proficiency [7,8]. Other countries appear to be experiencing similar phenomena regarding the increased adoption of foreign language education. For instance, [9] reported that parents in the United Arab Emirates expressed positive attitudes towards international schools. Additionally, a study of parents in Egypt found that the participants hoped that enrolling their children in international schools would improve their social mobility due to the perceptions that students enrolled in these schools had higher socioeconomic statuses and that other Egyptian school systems were deteriorating [10]. Regarding concerns about L1 proficiency, Vietnamese parents expressed a preference for maintaining Vietnamese in the curriculum alongside English rather than adopting a monolingual educational policy [11]. These findings and the findings of Lehman [12] indicating an absence of formal written policy, suggest that educational policy makers shall better align the curricula used in monolingual schools with those used in international schools in terms of content and quality in order to address parents' needs and concerns.

Public schools in Saudi Arabia are managed by the Ministry of Education (MoE) and are free of charge. Private schools are managed independently and require tuition fees. Many private schools in Saudi Arabia offer two educational pathways: a national programme and an international programme. National programmes at private schools are similar to those offered by public schools but focus more intensively on subject specialisation, such as English and applied science (mathematics, physics, chemistry). While teachers in the national programme have access to supplementary English textbooks, Arabic is the primary language of instruction. Conversely, international programmes typically adopt the curriculum of a specific country and are taught in that country's official language (e.g., English or French). Most English international programmes employ curricula from the United States (US) or United Kingdom (UK) for all courses, which are taught in English, as well as

Arabic and Islamic studies, which are taught in Arabic. In the past, these international programmes and schools catered to English-speaking expatriate communities living in Saudi Arabia; however, along with the expansion in the use of English in local labour market in Saudi Arabia, an increasing number of Saudi and Arab families have begun enrolling their children in international schools.

Subject to approval from the MOE, international schools are permitted to add or modify courses and adjust the number of subjects/courses. Thus, private schools often begin teaching English earlier and offer more subjects in English than public schools. Despite the expansion of English instruction in public schools, enhanced exposure to English in private schools has improved second language (L2) proficiencies among private school students beyond those of their peers in public schools. These schools and programmes seem to be attracting an increasing number of parents.

Despite the growing number of Saudi students registered in international schools in Saudi Arabia, to the best of our knowledge, no previous investigation has explored Saudi parents' attitudes towards these schools. Therefore, this study investigates parents' attitudes towards enrolling their children in international schools in an attempt to address a gap in the literature that is expected to broaden our understanding of the factors that encourage parents to enrol their children in different school systems, including returning expat parents who lived with their children in non-Arab countries for years. The study thus set out to answer the following research questions:

• Do Saudi parents consider English-language proficiency an educational need or luxury?

• To what extent do Saudi parents consider their children's language-of-instruction preferences when choosing educational institutions?

• Do Saudi parents consider international schools superior to Arabic schools that offer international programmes? Why?

Answering these questions could help understanding Saudi parents concerns regarding the potential linguistic influence of English on students L1. It may also help policy makers in understanding parents needs and fears regarding students' enrolments in international schools.

## 2. Literature review

This section reviews the history of English teaching in the Saudi education system, as well as the attitudes of Saudi parents' researchers, followed by the contextual and linguistics challenges in the Saudi schools.

### 2.1 English in Saudi schools

The teaching of English as a foreign language (EFL) was first introduced in Saudi Arabia in 1928 [13,14]. Since then, multiple changes, modifications and improvements have been made to EFL curricula in Saudi Arabia, especially in the last three decades, as part of an educational expansion initiative intended to support economic, commercial and industrial development [15]. Initially, the Saudi government opposed teaching English to public elementary school students (Grades 1–6) based on the assumption that it would negatively affect their L1 acquisition [16]. This policy lasted until 2010, when English was introduced in public elementary schools beginning in Grade 4. In 2021, English became a core course in public schools, with instruction beginning in Grade 1 [17].

Driven by labour market requirements, English is increasingly the dominant language of instruction in Saudi universities, where science programmes, such as engineering and medicine, are taught in English, while arts and other programmes are taught in Arabic [16]. Even in Arabic-language programmes, students are required to study EFL as elective or orientation courses. The emphasis on English-language education at the university level is attributed to English's status as an international language spoken widely by native and nonnative speakers [18]. While some have proposed that the MOE consider establishing bilingual programmes (i.e., Arabic and English) in public schools [19], others argue that the level of English proficiency required by labour market does not warrant such widespread adoption of English within

the public curriculum and warn that doing so may have unpredictable linguistic, social and cultural consequences. Indeed, Al-Jarf [1] points to a lack of linguistic policies and language planning in Saudi Arabia as a factor in slow Arabisation processes and low Arabic publishing output, an indication that the dominance of English in higher education poses a serious threat to Arabic. This increasing adoption of English-based instruction in Saudi universities seems to have shaped many parents' decisions to enrol their children in international schools, or "international schooling" as in [20].

## 2.2 Saudi Arabians' attitudes towards English

Successful English acquisition requires that an English-language education be perceived as beneficial for social and psychological reasons. Notably, Al-Zahrani [21] found that a considerable number of learners would not study English if given the choice, and Al-Seghayer [22] noted that many of the Saudi students who participated in the study believed that learning English was beyond their capability. Other studies, however, have shown generally positive attitudes towards English among Saudi learners. The vast majority of Al-Jarf's [1] student participants viewed English as the language of science, research and technology, which afforded it a superior status. More broadly, Faruk's [23] participants reported positive attitudes towards English and attributed their opinion to the belief that communicating in English is vital for the future of the country and a requirement in various economic, administrative and commercial domains. These studies suggest a shift in the general status and reputation of English in the Saudi context. More recent research has explored attitudes towards English-language education preferences. Specifically, Al-Jarf [24] found that 70% of 300 Saudi mothers expressed a desire to enrol their children in kindergartens that teach English. Moreover, beyond EFL instruction, Al-Matrafi [25] found that Saudis held positive attitudes towards teaching English culture and literature more broadly. The above studies suggest the Saudi students' motivation to acquire English is extrinsic rather than intrinsic. Additionally, the attitudes towards English and the motivation to learn it seem to be influenced by a number of challenges discussed in the remainder of this chapter.

Thus, teaching English speaking skills to Arab students has always been a challenging task for EFL instructors due to its status as a foreign language [26]. However, the question raised is whether children should or need to be enrolled in international schools. In addition, does parents recognition of the importance of English for their children and their positive attitude towards it implement a similar attitude towards international schools?

## 2.3 Classrooms

One of the main challenges in Saudi Arabian public schools is the number of students in classes. The number of students in one class can reach 50 in public schools in Saudi Arabia, whereas the maximum in international schools is 25. Smaller classes may provide greater learning opportunities for students [27,28], and their social and academic involvement can be affected by class size, with smaller classes offering fewer opportunities for misbehaviour, as learners are usually engaged in intensive social interaction [29]. Conversely, large classrooms can be difficult for teachers to manage [30], and instructors may struggle to support their students' language proficiency development [31], in part because there is insufficient capacity to afford each child the required individual attention or practice opportunities [32]. In these cases, weak or shy students may be neglected [33].

Small classes allow teachers to give more time to students as individuals according to their needs [34]. The time-consuming nature of classroom management in large classes can reduce the time spent on teaching and negatively impact learning momentum [34,35], and large classes tend to offer fewer opportunities for high-quality teaching or create a classroom environment that is not conducive to learning [36,37]. As mentioned above, students have fewer opportunities for L2 oral practice in large classes [38]. Despite recommendations that the number of students in a class should not exceed 12–30 students [39,40], Some studies contradict the suggestion to minimise class sizes based on findings that student performance can be better in larger classes because large classes are usually assigned to experienced teachers, while new teachers are given smaller classes with lower-proficiency students [41].

Another challenge in some Saudi public classrooms is the shortage of teaching equipment and facilities. Resources such as posters, language software, laboratories and flashcards are often absent [42], out of order or poorly maintained [43]. The absence of technology and other teaching tools extends to English classrooms in Saudi schools [44]. International schools on the other hand are usually provided with quality facilities, and consequently has a better level of belief [45].

### 2.4  Exposure to a second language (L2)

Despite growing interest in L2 education, exposure to English outside educational institutions tends to be low in Saudi Arabia, especially in general conversation, due to its continued status as a foreign language [22]. The lack of exposure to English in daily interactions is a significant barrier for Arab learners who wish to achieve higher levels of proficiency [16,46]. Even in the classroom, the L1 interferes with L2 acquisition [41]. Although research indicates that Arabic is widely used in the English classroom [e.g., 47], many studies suggest intensive exposure to the L2, combined with intensive communication in the school's context, to enhance L2 acquisition [26]. Various factors contribute to L1 use in the classroom, including teachers' lack of competence, low confidence in their English skills or the choice to make their job easier [47]. However, using L1 as the language of instruction decreases L2 practice and communication [48], which undermines L2 acquisition [42]. Alsuhaibani [49] encourages the facilitative use of Arabic in L2 classes.

### 2.5  Teachers, curricula and teaching styles

Additional issues have been observed in teaching styles and the English teaching curriculum in public schools in Saudi Arabia. In particular, English-language education in Saudi schools follows a teacher-centred pedagogical style characterised by a knowledge transmission strategy in which the teacher dominates the interaction [22,50]. Despite being outdated, these teaching approaches remain in use [44]. Activities in Saudi schools are not student-centred, and the passive role that students adopt in class means that they are usually quiet [42]. The lack of opportunity to take on more active roles in their learning deters students from developing their L2 proficiency [51–53]. Alnasser [44] and Fareh [42] both noted that Saudi pedagogy is outdated: syllabi and curricula are not revised to keep pace with contemporary teaching approaches, while teachers are not adequately trained in up-to-date teaching strategies.

## 3.  Methods

### 3.1  Research design

The study employed a mixed-methods design and collected quantitative and qualitative data from a sample of parents. A questionnaire was designed to elicit data on Saudi parents' attitudes towards enrolling their children in international schools. Student parents should provide us with accurate answers regarding their choices for choosing the schools in which the children are enrolled.

### 3.2  Participants

The study employed snowball sampling of parents across different regions of Saudi Arabia to ensure a representative study population and to attain a better overall understanding of Saudi parents' attitudes and opinions concerning the need for international schools. The target population is socially and economically diverse, and parents from different backgrounds and regions might have different perceptions of international schools or be subject to different social, geographic and economic factors that influence their opinions. Polling participants from different areas in Saudi Arabia also increased the chance of including families with experience living abroad for study or work from different Saudi backgrounds. Moreover, the different types of schools available in each region might affect parents' perceptions of international schools.

The participants comprised 76 parents, 52 male respondents (68.4%) and 24 female respondents (31.6%), with at least one child currently or previously enrolled in an international programme in Saudi Arabia. Of the 76 respondents, 93.4% lived with their partners at the time of the study, while 6.6% were single parents. Most (89%) were between 27 and 47 years of age. Almost all (around 99%) held at least a bachelor's degree, although 61% of the sample held a postgraduate degree (26.3% MA, 35.5% PhD). Collecting more data from less educated parents was not possible because most parents who register their children in international schools are well educated, but the sample did include respondents from a range of socioeconomic backgrounds based on annual income. Most respondents (61.4%) had lived abroad for at least one year, with most of these experiences occurring during postgraduate studies. In all, 51.3% of the respondents reported that they had enrolled at least one child in school in the country in which they lived at the time, while the rest did not have children while they were abroad.

### 3.3 Data collection

This study employed a questionnaire for data collection designed and administered to the target population through Google Forms. A questionnaire is a tool commonly used in sociolinguistics research to measure attitudinal, factual, behavioural and statistical dimensions [54]. Questionnaires have also been widely identified as the most effective instrument applied in bilingual education research [55]. According to Curle and Derakhshan [56], its 'extensive use originates from the many advantages that questionnaire use brings to researchers in terms of its practicality, economy, feasibility, time, efficiency, versatility, ease of construction and data analysis' (p. 3). Consequently, collected qualitative and quantitative data are detailed and insightful.

#### 3.3.1 Survey design.
Consistent with Dörnyei's [54] view that questionnaire statements are used to assess people's beliefs, interests, opinions, attitudes and values, the questionnaire items were designed using Google Forms to capture Saudi parents' opinions and perceptions of international schools in Saudi society and of enrolling their children in these schools (see S1 Appendix). More specifically, the questionnaire included 30 items that focused on how perceptions of English instruction influenced participants' opinions and perceptions. The questionnaire included statements to which participants responded using a 5-point Likert scale (1 = strongly agree, 2 = Agree, 3 = Neutral, 4 = Disagree, 5 = strongly disagree). Open-ended questions were also included to facilitate more detailed responses.

In addition to the two-item formats, the questionnaire was divided into two sections. The first section included six questions of demographic information from the respondents to provide insights into the backgrounds of the participating parents including respondent's' legal capacity, age, qualification, marital status and financial income.

The other subsections pertained to parents' beliefs about the benefits and challenges of international schools. To enhance accessibility and ensure clarity, the questionnaire was developed and administered in Arabic, thus eliminating any language barriers that might hinder comprehension or participation. The questionnaire underwent a critical review prior to distribution and was scrutinised by three experts in the field, whose valuable input was incorporated to enhance its quality via rephrasing and clarifying some of the questions.

#### 3.3.2 Survey distribution and data management.
The questionnaire was then distributed online via WhatsApp to various individuals at universities and other educational institutions. We also asked the participants to distribute the questionnaire to others through a snowballing process to enhance the quality of the study and reach the highest number of participants. For the qualitative data, the respondents' answers to the open-ended questions were analysed thematically. Thematic analysis aims to identify patterns and meanings within data [57]. Hence our qualitative data are classified and analysed under four main themes implied in the responses of the participants: language, context and social norms, the features of school/program and career and higher education prospect.

### 3.4 Ethical considerations

The study addressed ethical considerations with care throughout. Ethical approval was obtained from the first author's university (No. ETHICS2067), ensuring that the research complied with ethical standards and guidelines. The anonymity of

all the participants was maintained and the information they provided was treated confidentially. The data collected were used solely for the purposes of this research and were accessible only to the researchers. Written consent was obtained. Before completing the questionnaire, it was clearly stated on the first page of the questionnaire that by moving to the next pages and completing the survey, participants were agreeing to participate in the study. All participants were adults who agreed by clicking on and moving on to the subsequent pages of the questionnaire. The survey produced quantitative and qualitative data. For the quantitative data, descriptive statistics were calculated with a focus on percentages.

## 4. Findings

The findings indicate that parents choose international programmes for their children for various reasons. Based on the qualitative data analysis, four major themes emerged: (1) language, (2) context and social norms, (3) school and programme features and (4) career and higher education prospects. Theme number one was related to the first question while themes two and three were related to research question number two. Theme four was related to research question three. Each theme is presented below in a separate section.

### 4.1 Language

Many parents expressed the belief that international school programmes help strengthen learners' L2 competency from early childhood. One respondent stated that professional English was his primary reason for choosing an international school. Two other respondents noted that exposure to L2 aids learners in becoming bilingual and acquiring knowledge in both languages. Another respondent stated that the aim was to prepare his child to become 'extremely fluent' in English. Overall, the respondents expressed the belief that international schools offered intensive exposure to English, which helped their children in various domains. However, some respondents criticised the focus on English-only instruction in international schools and reported concerns that this could affect their children's L1 proficiency as well as community integration and cultural identity.

### 4.2 Context and social norms

Enrolling Saudi children in international schools seems to have stemmed from the children's needs to pursue their education which started in an English-speaking country. The analysis showed that 68.4% of respondents had spent at least one year studying or working abroad, while others intended to travel abroad to pursue their studies in the near future. The former described enrolling their children in international schools for continuity upon returning to Saudi Arabia, while the latter framed international school education as a preliminary step in preparing their children for life abroad. One respondent explained their choice of an international school based on the potential for sponsorship for the family to pursue postgraduate studies abroad, as it might help the transition to an English-speaking school when overseas. Another father attributed his choice to his wife's plan to pursue her PhD abroad. For these respondents, enrolling their children in international schools while still in Saudi Arabia was a proactive measure intended to ensure a smooth transition from public Arabic schools to international schools while abroad.

Respondents also explained their rationale for choosing international schools by referring to the fact that children understood English better than Arabic or could not speak Arabic at all after years of studying abroad. One respondent framed her children's attendance at an international school as a temporary decision intended to ease the transition from foreign schools to Saudi public schools. Similarly, another respondent considered a period of time at an international school to be an extension of what children had started abroad and an opportunity to minimise linguistic and cultural shock after studying abroad. These parents opted for international schools to minimise disruption: Their children had begun their education in Arabic schools, then started studying in English abroad in the absence of Arabic alternatives and had difficulties adapting abroad, so upon returning home, the respondents enrolled their children in international schools to avoid further disruption, despite the high cost.

Respondents also noted that their children's acquisition of English as an L1 while abroad made it difficult for some to learn *in* Arabic, thus motivating the choice of an international school upon returning home. One respondent described his child as 'monolingual' after overseas scholarship, albeit acknowledging the enrichment offered by international schools. Another respondent stated that after his children had spent three years in international schools – and lost their Arabic while living in Saudi Arabia – he decided to move them to Arabic programmes in private schools. Table 1 shows how the respondents described their children's levels of proficiency in both Arabic and English. Parents also mentioned excluding public schools as an option due to crowded classes and the potential for bullying.

The respondents also reported concerns about social and community-related pressure as factors that influenced their choice of schools, including high competition and the stress of keeping up with society. Specifically, some respondents expressed the belief that public schools were ''not up to the mark,'' including one whose choice was influenced by better learning opportunities for his son. This respondent described the international school as an 'extended family' and indicated 'that [he] was deprived of such opportunities' as a motivation. Moreover, some parents explained that they chose international schools because they cater to particular social classes and contexts. As one respondent explained, 'To be honest, the pressure of society is strange, asking about the school in which my son is enrolled'.

## 4.3 School and programme features

The respondents reported that they perceived international schools as offering higher-quality education than public schools and were attracted by the smaller class sizes. One respondent mentioned that international schools have good, strict policies regarding student dismissal compared to public schools.

Respondents also chose international schools based on whether they adopted a UK, US or Australian curriculum. As one respondent explained, these curricula are much better than those provided in public schools or even in national (Arabic) programmes in private schools. Respondents also described international schools as 'enriching' learning environments that focus intensively on student development in several areas, including language, science, self-development, leadership and public speaking. Respondents expressed the view that international schools teach students how to think, analyse and make decisions using cognitive, critical and deduction strategies. Respondents also praised international schools for meeting the minimum standards of knowledge transfer in addition to developing skills. Moreover, respondents perceived international schools as employing competent teachers and offering attractive coursebooks and materials. Many respondents also identified English instruction in science courses as a positive factor.

Notably, one respondent expressed scepticism about the regional curricula, stating that the teaching system does not reflect that used in the countries in which the curricula are originated. Another respondent agreed and claimed that not all international schools maintain high standards, with some recruiting poor-quality staff who apply ineffective teaching styles. Nonetheless, the participants concluded that even these international schools were better than other (public) schools. Indeed, many respondents described the quality of education available in public schools as poor and favoured international schools due to a focus on learning outcomes and innovative pedagogical approaches.

**Table 1. Parents' descriptions of their children's current levels of proficiency in Arabic and English.**

| Arabic | | English | |
|---|---|---|---|
| *Level* | *Percentage* | *Level* | *Percentage* |
| Excellent | 38.2% | Excellent | 59.2% |
| Above average | 21.1% | Above average | 28.9% |
| Average | 30.3% | Average | 10.5% |
| Weak | 7.9% | Weak | 1.3% |
| Very weak | 2.6% | Very weak | 0% |

## 4.4 Career and higher education prospects

Many respondents expressed the belief that international schools would better prepare their children for future studies abroad, which they anticipated given the lack of higher education opportunities at home. Respondents also mentioned that international programmes help students with obtaining the English proficiency they will need upon joining university and later in the labour market. Respondents also identified the need for L2 acquisition to begin at an early stage to facilitate the pursuit of graduate studies at English-language institutions at home or abroad. In another evidence for the extrinsic motivation which parents have about international schools, one respondent described English as 'the language of the era'. Many respondents expressed concern about their children possibly failing to achieve the English language proficiency required by some universities in Saudi Arabia and believed that enrolling their children in international schools might help overcome this challenge. When asked which language they considered more relevant to students' future learning and recruitment opportunities, an overwhelming 93.4% of respondents indicated that English is more important than Arabic. In addition to general English skills, which could be learned in contexts other than international schools, respondents identified English scientific vocabulary acquisition as a prerequisite for studying abroad.

## 4.5 Additional factors

In addition to the previously mentioned reasons for opting for international schools, parents also mentioned other motives, which we list in this section. Respondents with a child or children enrolled in international schools responded overwhelmingly positively (72% 'Yes', 28% 'No') when asked whether they would enrol all their children in an international school if it were financially viable. Some respondents indicated additional relevant factors, including the quality of education and their children's willingness and ability to attend and perform well at an international school. Respondents who had not enrolled their children in international schools identified several factors that influenced their decision, including individual differences, a child's reluctance, a weak programme focus on Arabic, the potential loss or weakening of a child's L1, the preservation of cultural identity, concerns related to values and religion, a child's lack of English skills, health issues, a perceived decline in educational quality in international schools and programmes, increasing costs, transportation barriers, schools being too business oriented and parents' limited exposure to language and sciences, which might make the effort of following up difficult.

## 4.6 Quantitative data

Most respondents (61%) expressed the belief that the student community in international schools was outstanding. Moreover, 67.2% felt that their children were outstanding because they could study in international schools. Most (80%) were interested in choosing international schools because they adopted curricula from scientifically advanced countries. More than one-third of respondents (38.2%) reported that some or all of their children studied at international schools because they could not speak Arabic and had adapted to studying in international schools in Saudi Arabia. Notably, data indicate that English acquisition was not a primary factor for many respondents: Only 9.2% indicated that they had chosen international schools to develop their children's proficiency in English, and most (78.9%) reported a dual interest in their children developing language *and* science skills.

## 4.7 Reputation of international school graduates

The respondents indicated a strong belief that international schools and their graduates have positive reputations. Most (72.4%) stated that they preferred international schools due to their strong reputation in the Saudi community, with 63.1% specifying that international school graduates are recognised as outstanding and distinctive. Furthermore, 72.3% of respondents indicated that they had enrolled their children in international schools based on the expectation that they would have a competitive advantage over graduates from other schools. The vast majority of respondents (85.5%)

reported the expectation that international school graduates outperform their peers from other schools regarding L2 proficiency, and 75% of respondents expressed the same belief regarding the sciences. Regarding academic competition, 64.5% of respondents anticipated that international school graduates would have a better chance of admission to university. Most parents (65.8%) reported that they chose international schools because they promote better school–parent relationships and are housed in better-quality buildings. When asked whether they believed that the educational outcomes of international schools were superior due to high tuition fees, most respondents reported agreeing 39.5%, or being neutral 39.5 on this point (Table 2).

## 5. Discussion

Do Saudi parents consider English-language proficiency an educational need or luxury?

To what extent do Saudi parents consider their children's language-of-instruction preferences when choosing educational institutions?

Do Saudi parents consider international schools superior to Arabic schools that offer international programmes? Why?

The findings of this study agree with those of previous research reporting that Saudi parents are increasingly sending their children to international schools. In the present study, most respondents were in favour of English as the primary language of classroom education [23–25] which is related to the first research question' Do Saudi parents consider English-language proficiency an educational need or luxury? Many of respondents' reasons for choosing international schools for their children were driven by external motivations, such as the high quality of education, better chances for success in university education and equipping their children with the English competency required in the labour market which is related to the second research question' To what extent do Saudi parents consider their children's language-of-instruction preferences when choosing educational institutions? Also, the answer of the third research question reveals that prestige and social status influence parents' decisions to enrol their children in international schools. This finding is in line with [58]. In Saudi society, individuals place considerable importance on how they are perceived by others, particularly within their peer groups in the workplace. Some parents indicated that they do not want to be perceived unfavourably by their peers based on the schools their children attend. Notably, most respondents identified as women, which suggests an avenue for future study regarding the intersection of gender, class and status in the Saudi context.

Parents also tend to view international schools modelled on US or UK systems as having better and stricter educational policies, which result in a higher quality of curricula and learning environments. Indeed, international schools are well known for providing professional educational environments and curricula developed by renowned educationalists with extensive experience in the field [44]. The facilities in the schools are another advantage that international schools have over public schools [42,59]. These factors may influence parents' assumptions that these schools are well equipped to provide their children with better futures by, for example, granting preferential access to the labour market. Moreover, experience studying abroad significantly influenced respondents' perceptions of international schools: They wanted their children to follow in their footsteps and attain a prosperous future in the labour market.

**Table 2. Perceived link between the excellence of international schools and high costs.**

| Level | Percentage |
|---|---|
| Strongly agree | 6.6% |
| Agree | 32.9% |
| Neutral | 39.5% |
| Disagree | 18.4% |
| Strongly disagree | 2.6% |

Another major factor that influences whether Saudi parents enrol children in international schools is related to language proficiency and educational experience. If their children were born or raised abroad and speak English as their L1, parents may fear disengagement in the classroom and be concerned about their child's lack of Arabic proficiency negatively affecting their confidence and self-esteem [60]. In these cases, returning expatriates may consider international schools the only reasonable option. Notably, anxieties about Arabic proficiency also influence the decision to enrol children with Arabic as their L1, and some respondents expressed concerns about international schools inhibiting Arabic language development.

Previous studies have revealed a growing trend of Saudi parents sending their children to international schools. This preference is driven by various factors, including the perceived superior educational policies derived from foreign school systems, the expectation that their children will have better career prospects and parental study abroad experiences. Moreover, the appeal of international schools has also been attributed to their curricula, educational environments and the prestige they offer to students and their families. However, it should be noted that there is debate concerning the legitimacy of international schools as lucrative institutions, which could affect their perceived status and prestige as well as the credibility of their educational outcomes [61]. This issue warrants further exploration in the Saudi educational context. It should be noted that the findings of the study were based on a small sample of participants. Despite our efforts to collect data from as many parents as we could, it was not possible to collect more data due to low turnover of the online questionnaire. Hence, the findings of the study are only interpreted as indicative of the Saudi parents' perceptions.

## 6. Conclusion

International schools play a significant role in supporting multicultural societies by promoting cultural understanding among students and preparing them to engage fully in culturally diverse contexts. Parents appear to perceive these institutions as offering diverse and inclusive learning environments while equipping students with the skills and knowledge necessary to navigate an increasingly interdependent and multicultural society. In addition, parents' decision to choose international schools for their children reflects a realisation of the significance of English. International schools have significant advantages and play a significant role in shaping the educational landscape for several reasons: preparation for higher education and enhanced career opportunities. One of the limitations is the limited number of parents who contributed in this study. Therefore, future research can include wider number of participants in the Saudi context. In addition, conducting similar studies in various countries may help in recognising whether parents in different communities/countries share similar attitudes in order for findings to be generalised. Finally, policy makers as in [62], can prioritise investment in international schools as potential institutions in initiating technologies in education.

## Supporting information

**S1 Appendix. Questionnaire.**
(DOCX)

## Author contributions

**Conceptualization:** Fahad Almulhim, Ali Alsaawi.

**Data curation:** Fahad Almulhim, Mohammad Almoaily.

**Formal analysis:** Fahad Almulhim, Hamza Alshenqeeti, Ali Alsaawi.

**Investigation:** Ali Alsaawi.

**Methodology:** Hamza Alshenqeeti.

**Project administration:** Nesreen Alahmadi.

**Software:** Mohammad Almoaily.

**Supervision:** Hamza Alshenqeeti.

**Validation:** Mohammad Almoaily, Nesreen Alahmadi.

**Visualization:** Fahad Almulhim.

**Writing – original draft:** Fahad Almulhim.

**Writing – review & editing:** Fahad Almulhim, Nesreen Alahmadi.

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
