## [Editor Report · Decision Letter 0]

4 Dec 2024

PONE-D-24-51475A need or a luxury? A study of parents’ attitudes towards choosing international schools for their children from a linguistic perspectivePLOS ONE?

Dear Dr. Almulhim,

Thank you for submitting your manuscript to PLOS ONE. After careful consideration, we feel that it has merit but does not fully meet PLOS ONE’s publication criteria as it currently stands. Therefore, we invite you to submit a revised version of the manuscript that addresses the points raised during the review process.

We look forward to receiving your revised manuscript.

Kind regards,

Anandhan Hariharasudan, PhD

Academic Editor

PLOS ONE

Journal Requirements:

3. In the online submission form, you indicated that the questionnaire responses are available upon request.

Additional Editor Comments :

The supporting documents are missing in this submission.

The paper cannot be considered for review until it submitted with all required files (Questionnaire, Data set, etc.).
---

## [Author Response · Author response to Decision Letter 1]

15 Mar 2025

Thanks for your valued comments. The manuscript has now been revised, taking editor’ comments into account. Two versions are now submitted, the original copy, and a revised copy with tracked changes. Referencing style in the revised version is now according to PLOS ONE referencing style. Also, the ethics part was revised to clarify the ethical procedure during data collection. Also, a copy of the questionnaire is uploaded.

---

## [Decision Letter · Decision Letter 1]

15 Oct 2025

PONE-D-24-51475R1A need or a luxury? A study of parents’ attitudes towards choosing international schools for their children from a linguistic perspectivePLOS ONE?

Dear Dr. Almulhim,

Thank you for submitting your manuscript to PLOS ONE. After careful consideration, we feel that it has merit but does not fully meet PLOS ONE’s publication criteria as it currently stands. Therefore, we invite you to submit a revised version of the manuscript that addresses the points raised during the review process.

We look forward to receiving your revised manuscript.

Kind regards,

Anandhan Hariharasudan, PhD

Academic Editor

PLOS ONE

Journal Requirements:

Additional Editor Comments (if provided):

The manuscript presents a relevant and potentially valuable topic; however, it requires substantial revision to meet publication standards. The research questions lack depth and insight, and the study’s novelty and distinction from existing literature are not clearly articulated. The literature review does not sufficiently justify or support the identified research gap, and several sections suffer from semantic redundancy due to repetitive phrasing. The conclusion reads more like a list than a cohesive synthesis, lacking logical connections between ideas. The sample size is too limited to generalize the findings, and methodological details—including sampling regions, inclusion criteria, data collection process, and survey validation—are inadequately described. The absence of the questionnaire in the appendix further weakens the study’s transparency. Moreover, the discussion does not critically engage with the findings, omits study limitations, and fails to offer practical implications or future research directions. The writing requires major language refinement to improve clarity, grammatical consistency, and coherence, with particular attention to verb tenses, pronoun references, and sentence transitions. The introduction and certain subsections need to be more concise, incorporating recent developments such as the growing influence of social media on education in Saudi Arabia. Overall, the paper must undergo major revision to enhance its theoretical contribution, methodological rigor, linguistic quality, and overall scholarly value.

Reviewers' comments:

Reviewer's Responses to Questions

**Comments to the Author**

Reviewer #1: (No Response)

Reviewer #2: (No Response)

2. Is the manuscript technically sound, and do the data support the conclusions?

Reviewer #1: (No Response)

Reviewer #2: Partly

3. Has the statistical analysis been performed appropriately and rigorously?

Reviewer #1: (No Response)

Reviewer #2: No

4. Have the authors made all data underlying the findings in their manuscript fully available?

Reviewer #1: (No Response)

Reviewer #2: Yes

5. Is the manuscript presented in an intelligible fashion and written in standard English?

Reviewer #1: No

Reviewer #2: No

Reviewer #1: 1. The abstract of the paper is written well.

2. The statistical analysis and the illustrations are adequately done.

3. However, research questions framed are not so insightful.

4. The study's novelty is not mentioned. How does the study differ from the existing one?

5. The literature review section does not say anything in support of the research gap identified and addressed.

6. Semantic redundancy is created by the repetitive use of certain phrases "global perspectives," "interconnected world," "multicultural society," "cultural exchange" "Significant role" and "pivotal role." This needs to be addressed by the researchers.

7. The concluding statement of the paper says --- “global communication, preparation for higher education, enhanced career opportunities, the facilitation of mobility and cultural exchange” — provide a significant advantage for English as a medium of instruction. It reads more like a bullet point list than an integrated sentence. The logical links between English and each of these points are asserted but not explained.

8. The sample of the study was very small (quantitative and qualitative data through a 30-item questionnaire to which 76 parents responded) and the authors try to claim to address the needs of the students of Saudi Arabia.

9. Instead of providing one-side view, the authors can acknowledge challenges and try to provide possible ways to resolve it.

10. The authors can avoid using vague transitions in sentences. Try to use bridging sentences.

11. The authors ought to eliminate generalized claims and think of adding specific examples.

12. The authors need to remove the unsubstantiated references found in support of their argument. Proper citation is the way out.

13. The Questionnaire used during the pre-study period was not provided in the appendix.

14. The standard of English needs to be improved.

15. Grammatical Errors need to be addressed:

Verb Tense Consistency:

• “Another major reason for enrolling children in international schools is if they were born or raised abroad…”

→ “is if they were” → awkward and inconsistent; consider parallel tense.

Incorrect or vague pronoun reference:

• “These are important for the future of the children.”

→ “These” refers to “quality of the curricula and the environment” but is too vague here.

Lack of Clarity in the Final Sentence found in the Discussion:

• “This could affect their perceived status and prestige, as well as the credibility of their educational outcomes.”

→ Ambiguous: whose perceived status — the schools or the students?

16. Scope for further is research is not given.

17. Hence it is perceived that the study lacks many important and crucial elements in support of the claims made by the authors.

18. Therefore, the paper needs to be sent for major revision.

19. The following article can be cited in the revised manuscript as there are some relevant ideas in support main argument of the research paper in the section: 2.3 Impact on Human Skills and Education:

Asokan Vasudevan, K. T. Tamilmani, Suleiman Ibrahim Mohammad, V. S. Sridheepika, N. Raja, Eddie Eu Hui Soon, Sriganeshvarun Nagaraj, and Ahmad Samed Al-Adwan. (2025). Voices of Concern: Contemplations on Socioeconomic, Ethical, and Developmental Impacts of AI. Applied Mathematics and Information Sciences: An International Journal, 19 No.1, 221-232. ISSN 2325-0399 (online) https://www.naturalspublishing.com/Article.asp?ArtcID=29328

Reviewer #2: - The topic is interesting, but the study itself does not add much to the reader’s knowledge and does not provide future directions to make the results valuable

-Language used in the article should be revised for better choice of words and punctuation. -The title implies more outcomes than the paper provides.

Introduction:

The introduction provides valuable information to help readers understand the educational system in Saudi Arabia; however, it would be more helpful if it were more concise.

The studies on “Saudi Arabians’ attitudes towards English” are outdated and may not accurately reflect the current status in Saudi Arabia, given the significant changes in the country and the shift toward extensive social media use over the last few years. This change should be reflected as well, and might add a great understanding to the topic

2.4. Classrooms: It would be helpful to provide the reader with the actual number of students in public vs private schools in Saudi Arabia to be able to decide on the difference and to understand if the class size is appropriate or not. In addition, whether this trend of choosing private schools over public ones has increased over time. Again, it would help the reader to have this part written in a summarized, concise way instead of mentioning different studies with different results

Sample: Which regions of Saudi Arabia were included in the study? What were the inclusion and exclusion criteria? The specific results on the sample should be added under

the results section.

Data collection:

The data collection section should be revised to provide a clear description of the process. It is mentioned that “The questionnaire was then distributed to various individuals at universities and other educational institutions. We also asked the participants to distribute it to others through a snowballing process to enhance the quality of the study.” Why were individuals at universities directly targeted, and wouldn't this create bias? And what is meant by educational institutions? Do you mean schools?

This section has information regarding developing the survey. It is recommended to separate the information into another section. Regarding the survey, it needs to be described more straightforwardly; it is not clear what the survey questions were. Perhaps it would be helpful to include them. More importantly, the survey’s validity was not mentioned, and no piloting was done.

Findings:

This section is difficult to follow; it would benefit from being more focused to aid the reader's understanding.

In sections 4.1 and 4.2, the percentages of parents providing information are not included, which does not accurately represent the value of the results.

Discussion:

The results of this study are not surprising and may not add to our knowledge unless the authors apply their results to our reality. For example, you use the results to provide suggestions to improve the educational system.

Limitations of the study are not mentioned

.

Reviewer #1: No

Reviewer #2: No

---

## [Author Response · Author response to Decision Letter 2]

16 Dec 2025

Dear editor and reviewers

Thanks for your valued comments.

The revised manuscript is attached along with the rest of required files.

Many thanks

---

## [Decision Letter · Decision Letter 2]

11 Jan 2026

PONE-D-24-51475R2A need or a luxury? Parents’ attitudes towards international schools from a linguistic perspectivePLOS One?

Dear Dr. Almulhim,

The manuscript requires **minor revision** to improve clarity and methodological consistency. The abstract should clearly state the mixed-methods design, include key numerical results, and specify both quantitative and qualitative procedures. The literature review needs updating with recent studies to enhance relevance. The methodology section should correct demographic details, clearly explain the qualitative component, and separately report demographic and Likert-scale items. While quantitative findings are well presented, qualitative results are missing and should be reported with thematic analysis and aligned to the research questions. The discussion and conclusion should be revised to explicitly justify each research question based on the study’s findings.to improve clarity and methodological consistency. The abstract should clearly state the mixed-methods design, include key numerical results, and specify both quantitative and qualitative procedures. The literature review needs updating with recent studies to enhance relevance. The methodology section should correct demographic details, clearly explain the qualitative component, and separately report demographic and Likert-scale items. While quantitative findings are well presented, qualitative results are missing and should be reported with thematic analysis and aligned to the research questions. The discussion and conclusion should be revised to explicitly justify each research question based on the study’s findings.to improve clarity and methodological consistency. The abstract should clearly state the mixed-methods design, include key numerical results, and specify both quantitative and qualitative procedures. The literature review needs updating with recent studies to enhance relevance. The methodology section should correct demographic details, clearly explain the qualitative component, and separately report demographic and Likert-scale items. While quantitative findings are well presented, qualitative results are missing and should be reported with thematic analysis and aligned to the research questions. The discussion and conclusion should be revised to explicitly justify each research question based on the study’s findings.to improve clarity and methodological consistency. The abstract should clearly state the mixed-methods design, include key numerical results, and specify both quantitative and qualitative procedures. The literature review needs updating with recent studies to enhance relevance. The methodology section should correct demographic details, clearly explain the qualitative component, and separately report demographic and Likert-scale items. While quantitative findings are well presented, qualitative results are missing and should be reported with thematic analysis and aligned to the research questions. The discussion and conclusion should be revised to explicitly justify each research question based on the study’s findings.

plosone@plos.org. . </article>

We look forward to receiving your revised manuscript.

Kind regards,

Anandhan Hariharasudan, PhD

Academic Editor

PLOS One

Journal Requirements:

Additional Editor Comments (if provided):

The manuscript requires minor revision to improve clarity and methodological consistency. The abstract should clearly state the mixed-methods design, include key numerical results, and specify both quantitative and qualitative procedures. The literature review needs updating with recent studies to enhance relevance. The methodology section should correct demographic details, clearly explain the qualitative component, and separately report demographic and Likert-scale items. While quantitative findings are well presented, qualitative results are missing and should be reported with thematic analysis and aligned to the research questions. The discussion and conclusion should be revised to explicitly justify each research question based on the study’s findings.

Reviewers' comments:

Reviewer's Responses to Questions

**Comments to the Author**

Reviewer #1: All comments have been addressed

Reviewer #3: All comments have been addressed

2. Is the manuscript technically sound, and do the data support the conclusions?

Reviewer #1: Yes

Reviewer #3: Yes

3. Has the statistical analysis been performed appropriately and rigorously?

Reviewer #1: Yes

Reviewer #3: Yes

4. Have the authors made all data underlying the findings in their manuscript fully available?

Reviewer #1: Yes

Reviewer #3: Yes

5. Is the manuscript presented in an intelligible fashion and written in standard English?

Reviewer #1: Yes

Reviewer #3: Yes

Reviewer #1: The authors have made substantial revision of the paper. They have mentioned the novelty and the contribution of the paper clearly.

Reviewer #3: In abstract, a mixed-methods research design was employed; however, this is not explicitly mentioned within the abstract. Additionally, numerical values should be incorporated to substantiate the study’s findings. In addition, a survey for quantitative data collection is mentioned, the procedures for qualitative data collection are not specified.

The literature review is clearly articulated and supported by adequate references; however, recent studies (2025, 2024, 2023, 2022) should be incorporated to strengthen the relevance.

In the methodology section, the demographic details should be corrected as 52 male respondents (68.4%) and 24 female respondents (31.6%). Furthermore, no explanation is provided regarding the qualitative component of the study, which weakens the mixed-methods research method. Furthermore, mention number of demographic details and likert scale questions separately.

In the findings section, the quantitative results are clearly presented and interpreted. However, the qualitative findings are not addressed. It is recommended that qualitative results be presented under a separate subheading with proper thematic analysis. Correlate the likert scale questions and open ended questions to the research questions with proper justification.

In the discussion and conclusion sections, the research questions are not sufficiently justified in relation to the findings. These sections should be revised and justified to each research question.

.

Reviewer #1: No

Reviewer #3: No

---

## [Author Response · Author response to Decision Letter 3]

15 Feb 2026

Dear editor

Thanks for your valued comments. The authors have worked on these comments. A revised tracked manuscript, a clean version of the manuscript and a separate file containing responses to each comment is attached.

---

## [Decision Letter · Decision Letter 3]

9 Mar 2026

A need or a luxury? Parents’ attitudes towards international schools from a linguistic perspective

PONE-D-24-51475R3

Dear Dr. Almulhim,

We’re pleased to inform you that your manuscript has been judged scientifically suitable for publication and will be formally accepted for publication once it meets all outstanding technical requirements.

Kind regards,

Anandhan Hariharasudan, PhD

Academic Editor

PLOS One

Additional Editor Comments (optional):

The paper has been revised and improved significantly.

It can be considered for publication.

All the best.

Reviewers' comments:

Reviewer's Responses to Questions

**Comments to the Author**

Reviewer #1: All comments have been addressed

Reviewer #3: All comments have been addressed

2. Is the manuscript technically sound, and do the data support the conclusions?

Reviewer #1: Yes

Reviewer #3: Yes

3. Has the statistical analysis been performed appropriately and rigorously?

Reviewer #1: Yes

Reviewer #3: Yes

4. Have the authors made all data underlying the findings in their manuscript fully available?

Reviewer #1: Yes

Reviewer #3: Yes

5. Is the manuscript presented in an intelligible fashion and written in standard English?

Reviewer #1: Yes

Reviewer #3: Yes

Reviewer #1: The authors have given proper citation for the references. The Questionnaire used during the pre-study period was provided in the appendix. The standard of English has been improved substantially. Scope for further is research is given. Now the article in the present form has reached the level of publishable article.

Reviewer #3: (No Response)

.

Reviewer #1: No

Reviewer #3: **Yes:**Saranraj LoganathanSaranraj LoganathanSaranraj LoganathanSaranraj Loganathan

---

## [Editor Report · Acceptance letter]

PONE-D-24-51475R3

PLOS One

Dear Dr. Almulhim,

I'm pleased to inform you that your manuscript has been deemed suitable for publication in PLOS One. Congratulations! Your manuscript is now being handed over to our production team.

Kind regards,

on behalf of

Dr. Anandhan Hariharasudan

Academic Editor

PLOS One